# Look-ahead Reasoning with a Learned Model in Imperfect Information Games

**Ondřej Kubíček**
Artificial Intelligence Center, FEE
Czech Technical University in Prague
Czech Republic
kubicon3@fel.cvut.cz

**Viliam Lisý**
Artificial Intelligence Center, FEE
Czech Technical University in Prague
Czech Republic
viliam.lisy@agents.fel.cvut.cz

## Abstract

Test-time reasoning significantly enhances pre-trained AI agents' performance. However, it requires an explicit environment model, often unavailable or overly complex in real-world scenarios. While MuZero enables effective model learning for search in perfect information games, extending this paradigm to imperfect information games presents substantial challenges due to more nuanced look-ahead reasoning techniques and large number of states relevant for individual decisions. This paper introduces an algorithm LAMIR that learns an abstracted model of an imperfect information game directly from the agent-environment interaction. During test time, this trained model is used to perform look-ahead reasoning. The learned abstraction limits the size of each subgame to a manageable size, making theoretically principled look-ahead reasoning tractable even in games where previous methods could not scale. We empirically demonstrate that with sufficient capacity, LAMIR learns the exact underlying game structure, and with limited capacity, it still learns a valuable abstraction, which improves game playing performance of the pre-trained agents even in large games.

## 1 Introduction

Strategic reasoning and planning are key components of human intelligence, encompassing our ability to reason about possible outcomes of actions in complex situations, often with incomplete information and uncertain consequences. Although humans navigate such decision-making naturally, replicating this process in artificial intelligence remains a fundamental challenge. Games, with their well-defined rules and yet complex strategic landscapes, serve as ideal testbeds for developing and evaluating AI planning and reasoning methods (Mnih et al., 2015; Silver et al., 2018; Perolat et al., 2022; Schultz et al., 2025).

In perfect information games like Chess, Go or Shogi, look-ahead search algorithms as Minimax and Monte Carlo Tree Search (MCTS) have achieved superhuman performance by systematically exploring future states (Russell & Norvig, 2003; Silver et al., 2018). These methods traditionally rely on access to game rules to implement state transitions in the search. MuZero demonstrated that an agent can learn a model of the environment dynamics implicitly through interaction and use this learned model to perform MCTS planning, removing the dependency on explicit implementation of the rules (Schrittwieser et al., 2020).

Not requiring explicit pre-programmed representation of the environment greatly expands applicability of AI methods. A method that does not require explicit rules representation can be applied, for example, to create an AI opponent in a proprietary video game without access to its source code; to create AI opponents for a large database of games for an online game playing platform, where programming a suitable representation for each of them would be prohibitively expensive; or in a game design setting, where the game is repetitively modified without the need for a programmer to reflect the changes in the implementation.

However, extending the model learning paradigm to imperfect information games such as Poker or Stratego presents fundamental difficulties. Since players lack complete knowledge of the state of the game, theoretically sound look-ahead reasoning methods need to reason about the distribution

of all possible hidden states consistent with shared knowledge (Kovařík et al., 2023), which differs from the MCTS used by MuZero.

Our work aims to **enable look-ahead reasoning in two-player imperfect information games using a learned abstract model**, thereby **eliminating the need for explicit game rules** and also **enabling look-ahead reasoning in parts of the game intractable without abstraction**. Following the approach of Schrittwieser et al. (2020), we focus only on games without chance events, which is a large class that includes commonly used benchmarks like Dark Chess, Stratego, Battleship or Imperfect Information Goofspiel. This allows us to tackle the unique difficulties of learning an effective abstraction for imperfect information without conflating it with the separate challenges introduced by chance events. Our contributions are: We identify the necessary components that a learned model must capture to facilitate look-ahead reasoning under imperfect information. We introduce a training procedure to learn these components from sampled game trajectories. We demonstrate how tractably small, domain-independent abstractions can be learned concurrently with the model. Finally, we introduce a way to conduct look-ahead reasoning with the learned model.

Our empirical evaluation shows that in small games, given sufficient capacity, the strategies produced by look-ahead reasoning are less exploitable than those of concurrently trained Regularized Nash Dynamics (RNaD). Furthermore, in large games with shared knowledge consistent with over $10^{18}$ states for some decisions, the proposed look-ahead reasoning is still applicable in all decision points and significantly improves over RNaD, reaching up to 80% win rate in head-to-head play.

## 1.1 RELATED WORK

**Direct Policy Optimization**   One approach for computing strategies in large imperfect information games stores the strategy implicitly in neural network weights and directly optimizes this policy based on self-play traces. These methods include: policy-gradient algorithms with reward regularization, like Regularized Nash Dynamics (Perolat et al., 2021; 2022; Sokota et al., 2023; Masaka et al., 2025); training networks to approximate CFR results, like Deep CFR or DREAM (Brown et al., 2019; 2020; Steinberger et al., 2020); or iteratively training best responses to growing strategy pools, like PSRO (Lanctot et al., 2017; Vinyals et al., 2019). Critically, these approaches rely solely on the trained actor during gameplay and cannot refine decisions with additional test-time computation. Pre-trained agents without test-time reasoning are usually very exploitable (Wang et al., 2023; Lisý & Bowling, 2017), and adding test-time reasoning greatly improves their capabilities in games (Silver et al., 2016; Kubíček et al., 2024), and beyond (Snell et al., 2024). This paper enables adding test-time reasoning to policies created by direct policy optimization algorithms.

**Look-ahead reasoning**   Reasoning algorithms in imperfect information games, such as Counterfactual Regret Minimization (Zinkevich et al., 2007), iteratively improve player's policies by systematically iterating over all possible future action sequences in all possible (unobserved) states of the game. In large games, this requires either domain-specific abstractions, like Libratus (Brown & Sandholm, 2018) or depth-limited reasoning, like DeepStack, ReBeL, Student of Games, SePoT (Moravčík et al., 2017; Brown et al., 2020; Schmid et al., 2023; Kubíček et al., 2024). In either case, all these algorithms require explicit implementation of game rules to construct game trees and manage belief states. It limits their applicability when exact rules are unavailable, computationally prohibitive or if the amount of possible hidden states is too huge. Knowledge-limited subgame solving (Zhang & Sandholm, 2021; 2026; Liu et al., 2023) can reduce the complexity, but even this reduced state space remains intractable in the games we study here. In contrast, our approach does not require explicit game rules and automatically learns a tractably small abstraction of the game just from full traces of game play.

**Model learning**   In single-player settings, Dreamer, TD-MPC and subsequent works showed that learning models and generating artificial traces can match purely model-free approaches (Hafner et al., 2020; 2021; 2025; Hansen et al., 2022; 2024). MuZero demonstrated similar results in perfect information games, using learned models for reasoning during gameplay (Schrittwieser et al., 2020; Antonoglou et al., 2022). Our work extends these approaches to imperfect information games without chance. The model we learn is similar to TD-MPC (Hansen et al., 2022), but it trains additional components to account for the abstractions and more nuanced imperfect information search.

## 2 BACKGROUND

We define two-player zero-sum simultaneous move game as $\mathcal{G} = (\mathcal{N}, \mathcal{W}, w^{\text{INIT}}, \mathcal{A}, \mathcal{T}, \mathcal{R}, \mathcal{O})$ (Kovařík et al., 2022), where $\mathcal{N} = \{1, 2\}$ are the players, $\mathcal{W}$ is the set of world states in the game and $w^{\text{INIT}} \in \mathcal{W}$ is the initial world state. $\mathcal{A} = \Pi_{i \in \mathcal{N}} \mathcal{A}_i$ is the set of joint actions,. We use $\mathcal{A}(w) \subseteq \mathcal{A}$ to denote the set of joint legal actions in world state $w$. $\mathcal{T} : \mathcal{W} \times \mathcal{A} \to \mathcal{W}$ is the transition function and $\mathcal{R} : \mathcal{W} \times \mathcal{A} \to \mathbb{R}$ is the reward function, which corresponds to the reward of player 1. We use $\mathcal{R}_1(w, a) = \mathcal{R}(w, a)$ and $\mathcal{R}_2(w, a) = -\mathcal{R}(w, a)$ as rewards for player 1 and 2 respectively. $\mathcal{O} : w \times \mathcal{A} \times \mathcal{W} \to \mathbb{O}$ is the observation function. $\mathbb{O} = \mathbb{O}_0 \times \mathbb{O}_1 \times \mathbb{O}_2$ is the set of joint public and private observations. $\mathcal{O}$ can be factored as $\mathcal{O} = (\mathcal{O}_0, \mathcal{O}_1, \mathcal{O}_2)$, where $\mathcal{O}_0$ is a public part of observation and $\mathcal{O}_1, \mathcal{O}_2$ are private parts of observations for each player. Even though we focus on simultaneous-move games, this does not limit the generality, since any sequential-move game can be converted into a simultaneous-move game by adding fictitious moves for the non-acting player in each decision node.

History $h = w^0, a^0 \ldots a^{l-1} w^l \in (\mathcal{W}\mathcal{A})^* \mathcal{W}$ is a finite sequence of world states and actions, which starts in the initial world state $w^0 = w^{\text{INIT}}$ and for each timestep $t \in \{0, \cdots, l-1\}$ holds $a^t \in \mathcal{A}(w^t)$ and $w^{t+1} = \mathcal{T}(w^t, a^t)$. $\mathcal{H}$ is the set of all possible histories within the game. We use $h \sqsubseteq h'$ to denote that $h'$ contains $h$ as a prefix. We will use $h^{\text{INIT}} = w^{\text{INIT}}$ to denote the initial history. Each history $h$ ends with some world state $w^l$. We will sometimes use history $h$ in the game functions instead of the world state $w^l$. For example $\mathcal{A}(h) := \mathcal{A}(w^l)$ corresponds to the set of joint legal actions in the final world state of the history. Each player $i$ does not observe the whole world state at each timestep, but only observes public observations $\mathcal{O}_0$ and its private observations $\mathcal{O}_i$. This means that the player may not be able to distinguish between several different histories. We will use $s_i \in \mathcal{S}_i$ to denote the set of all histories consistent with the observations of the player $i$, which we will call the information set. $\mathcal{S}_i$ is the set of all the information sets of the player $i$. $s_i(h)$ is the information set that corresponds to history $h$. Similarly, the public state $s_0 \in \mathcal{S}_0$ is an information of an external player, which does not have private observations, so it contains all the histories consistent with public observations. Each public state contains one or more information sets for each player.

Strategy of player $i$ is a function $\pi_i : \mathcal{S}_i \to \Delta \mathcal{A}_i$ that maps each information set to a probability distribution over the actions. We will sometimes use $\pi_i(s_i, a_i)$ as a probability that player will play $a_i$ in information set $s_i$ if it follows the strategy $\pi_i$. $\pi = (\pi_1, \pi_2)$ is a joint strategy profile of both players. If $h \sqsubseteq h'$, then the reach probability of reaching history $h'$ from history $h$ under strategy profile $\pi$ is $P^\pi(h'|h) = \prod_{h''aw \sqsubseteq h'} \prod_{i \in \mathcal{N}} \pi_i(s_i(h''), a_i)$. We also use $P^\pi(h) := P^\pi(h|h^{\text{INIT}})$. Any reach probability can be factored into the contribution by each player $P^\pi(h|h') = \prod_{i \in \mathcal{N}} P_i^\pi(h|h')$. Expected utility of a history $h$ if all players follow strategy profile $\pi$ is $u_i^\pi(h) = \sum_{h \sqsubseteq h'a} P^\pi(h'|h) \mathcal{R}_i(h', a) \prod_{i \in \mathcal{N}} \pi_i(s_i(h'), a_i)$.

Best response against a strategy $\pi_i$ is a strategy $\pi_{-i}^{BR} \in BR_{-i}$, which maximizes the opponent's utility $u_{-i}^{(\pi_i, \pi_{-i}^{BR})}(h^{\text{INIT}}) \geq u_{-i}^{(\pi_i, \pi'_{-i})}(h^{\text{INIT}})$ for any $\pi'_{-i}$. We use a $-i$ here to symbolize the other player than $i$, which is a standard notation in games. If all players play a best response to each other, the resulting strategy profile is known as Nash Equilibrium $\pi^*$ (Nash, 1950; Kuhn & Tucker, 1951). In two-player zero-sum games, this is usually the sought after solution concept. As a metric to evaluate quality of a strategy, we use exploitability $\mathcal{E}(\pi_i) = u_{-i}^{(\pi_i, \pi_{-i}^{BR})}(h^{\text{INIT}}) - u_{-i}^{\pi^*}(h^{\text{INIT}})$, which is how much can opponent gain, when it plays best response as compared to the Nash equilibrium. In two-player zero-sum games, the exploitability is always nonnegative and is zero if and only if the $\pi_i$ is a part of Nash equilibrium.

## 3 LEARNING THE GAME MODEL

In perfect information games, players possess complete knowledge of the current game state represented by a history $h$. Consequently, search algorithms initiate from a single, known root state, simplifying the search. In contrast, imperfect information games (IIGs) grant players only partial observability through an information set $s_i$, which typically corresponds to multiple possible underlying world states. As established in prior works, approximating optimal strategies via lookahead reasoning in IIGs requires a more sophisticated approach than in perfect information settings (Kovařík et al., 2023; Moravčík et al., 2017).

Specifically, it is insufficient to restrict the reasoning to only those histories consistent with the player's $i$ information set. Instead, the reasoning must encompass all histories that share the public state $s_0$. This necessity arises because sound reasoning algorithms compute strategies for all players simultaneously, aiming for mutual best responses characteristic of an equilibrium. Consider this situation in two-player Poker: if player $i$ holds two Kings, their information set $s_i$ includes all histories consistent with this hand but with varying opponent hands. A reasoning restricted only to those histories would implicitly grant the opponent knowledge of $i$'s hand when computing opponent's strategy, leading to suboptimal strategies. Therefore, the look-ahead reasoning must operate over the broader set of histories consistent with public state to compute valid equilibrium strategies (Moravčík et al., 2017; Schmid et al., 2023; Kovařík et al., 2023; Milec et al., 2024).

We adopt a reinforcement learning paradigm where an agent learns from interaction with the environment. During training, we assume access to a game simulator capable of generating the whole game trajectories. During testing (gameplay), the agent receives only its own information set $s_i$ at each step and must rely entirely on its learned model to plan, without access to the simulator or explicit rules. This means that the agent does not use any domain-specific knowledge. at any point.

We propose a model inspired by MuZero (Schrittwieser et al., 2020) but adapted for the IIG setting, which requires additional structures necessary to model the imperfect information. Our model comprises three core learnable functions, parameterized by $\theta$:

- Representation function $\Lambda_\theta^I : \mathcal{S}_i \to \overline{\mathcal{S}_i}$. Maps a player $i$'s potentially high-dimensional information set $s_i$ to a fixed-size latent representation $\overline{s_i} \in \overline{\mathcal{S}_i}$.

- Dynamics function $\Upsilon_\theta : \overline{\mathcal{S}_1} \times \overline{\mathcal{S}_2} \times \mathcal{A}_1 \times \mathcal{A}_2 \to \overline{\mathcal{S}_1} \times \overline{\mathcal{S}_2} \times \mathbb{R} \times \mathbb{B}$. Given the latent representations for all players $(\overline{s_1}, \overline{s_2})$ and the joint action taken $(a_1, a_2)$, this function predicts the resulting next latent representations for both players, the immediate reward $r$ (e.g., for player 1), and a binary termination flag $l$. This models the joint evolution of the game across possible hidden states. We use $\mathbb{B} = \{0, 1\}$.

- Legal actions function $\Gamma_\theta : \overline{\mathcal{S}_i} \to \mathbb{B}^{|\mathcal{A}_i|}$. Predicts the mask of legal actions $\mathcal{A}_i$ available to player $i$ from their latent representation $\overline{s_i}$. This is crucial for constraining the look-ahead reasoning only to the feasible parts of the game.

Each training episode a game trajectory is sampled. This trajectory is $h = w^{\text{INIT}} a^0 \dots a^{l-1} w^l$, where $a^t = (a_1^t, a_2^t)$ is the joint action at step $t$. For any sub-history $h^t \sqsubseteq h$, the simulator provides the true information sets $s_i^t(h^t)$, legal actions $\mathcal{A}_i^t(s_i^t)$, and the reward $r^t$. The model is trained to predict these quantities through recurrent application of its components.

Starting from an initial latent state $\overline{s_i^{t,0}} = \Lambda_\theta^I(s_i^t)$, the dynamics function is unrolled for $k$ steps using the actual actions from the trajectory:

$$\overline{s_1^{t,k+1}}, \overline{s_2^{t,k+1}}, \overline{r^{t,k+1}}, \overline{l^{t,k+1}} = \Upsilon_\theta(\overline{s_1^{t,k}}, \overline{s_2^{t,k}}, a_1^{t+k}, a_2^{t+k})$$

Here, $\overline{s_i^{t,k}}$ is the predicted latent state after $k$ unrolls from step $t$, and $\overline{r^{t,k}}$ and $\overline{l^{t,k}}$ are the predicted reward and termination logit. The legal actions function predicts logits $\overline{A_i^t} = \Gamma_\theta(\overline{s_i^{t,0}})$ from the initial latent state.

The model parameters $\theta$ are optimized by minimizing a combined loss function over the trajectory:

$$\mathcal{L}_\theta^M = \sum_{t=0}^{l-1} \Bigg[ \sum_{i \in \mathcal{N}} \underbrace{\mathcal{L}_\theta^L(\overline{A_i^t}, \mathcal{A}_i^t)}_{\text{Legal Action Prediction}} \tag{1}$$
$$+ \sum_{k=1}^{l-t} \Big( \underbrace{\mathcal{L}_\theta^T(\overline{l^{t,k}}, \mathbb{I}[t+k=l])}_{\text{Termination Prediction}} + \underbrace{\mathcal{L}_\theta^R(\overline{r^{t,k}}, r^{t+k})}_{\text{Reward Prediction}} + \sum_{i \in \mathcal{N}} \underbrace{\mathcal{L}_\theta^D(\overline{s_i^{t,k}}, \Lambda_\theta^I(s_i^{t+k}))}_{\text{Latent State Prediction}} \Big) \Bigg]$$

where $\mathcal{L}_\theta^L$ and $\mathcal{L}_\theta^T$ are binary cross-entropy losses, while $\mathcal{L}_\theta^R$ and $\mathcal{L}_\theta^D$ are mean squared errors. The target for the dynamics loss is the latent representation of the actual subsequent information set.

Minimizing $\mathcal{L}_\theta^M$ trains the functions $\Lambda_\theta^I, \Upsilon_\theta, \Gamma_\theta$ to collectively act as a learned simulator. This learned model captures the necessary dynamics within the game, enabling test-time look-ahead reasoning algorithms (discussed in Section 5) to effectively plan over the required set of public states without recourse to the original game rules or simulator.

## 4 LEARNING THE ABSTRACT MODEL

A primary limitation of sound look-ahead reasoning in imperfect information games is the potential size of the public states, as the number of information sets consistent with public information may be exponential in the history length, making the look-ahead reasoning intractable (Moravčík et al., 2017; Schmid et al., 2023).

Although traditional abstraction techniques often rely on domain expertise or require offline enumeration and analysis of all information sets (Čermák et al., 2020; Kroer & Sandholm, 2018; Ganzfried & Sandholm, 2014; Brown & Sandholm, 2015; Bard et al., 2014; Johanson et al., 2013), we aim to learn domain-independent abstraction directly from the game experience during training. Our goal is to partition the vast space of information sets sharing a public state into a manageable number $L$ of abstract information sets, enabling tractable reasoning. Such an abstraction may contain imperfect recall as discussed in Section 7.

Consider Texas Hold'em Poker as an example. A player might hold any of the 1326 private hands, each corresponding to a different information set. Our abstraction aims to represent this multitude using only $L$ representatives, learned based on similarity within the training process rather than predefined rules.

We adapt the model from Section 3, but we will use mechanisms inspired by online clustering to limit the amount of information sets in each public state. We hypothesize that information sets behaving similarly, e.g. having similar optimal strategies or leading to similar future states, should be grouped. We formalize this using a function $\kappa : \mathcal{S}_i \to \mathbb{R}^K$, which maps any information set to a $K$-dimensional space, in which the clustering will be performed.

To satisfy the condition that each public state consists of at most $L$ information sets of each player, we split the representation function into two parts as shown in Figure 1. The public state representation $\Lambda_{i,\theta} : \mathcal{S}_0 \to \overline{\mathcal{S}_i}^L$ maps a public state $s_0$ to $L$ latent abstract information sets for a player $i$. The information set representation $\Lambda_{i,\theta}^I : \mathcal{S}_i \to \Delta\overline{\mathcal{S}_i}$ maps a real information set to the probability distributions on the abstract information sets provided by $\Lambda_{i,\theta}$. Crucially, despite $\Lambda_{i,\theta}^I$ providing a probability distribution, we enforce the many-to-one mapping, so we represent any real information set by the single abstract information set, corresponding to the highest probability. This representative will then be used for training dynamics $\Upsilon_\theta$ and legal actions $\Gamma_\theta$. We opted to this, so that the dynamics function constructs the search tree, which is compatible with look-ahead reasoning algorithms like Counterfactual Regret Minimization.

In order to perform the clustering, we require the same function $\kappa$ as for the real information sets. We introduce $\kappa_\theta : \overline{\mathcal{S}_i} \to \mathbb{R}^K$, which will be trained to predict this clustering property for each abstract information set. The dynamics function $\Upsilon_\theta$ and legal actions function $\Gamma_{i,\theta}$ are defined as in Section 3.

To learn the abstraction that approximates the proposed clustering based on $\kappa$, we train $\Lambda_{i,\theta}, \Lambda_{i,\theta}^I$ and $\kappa_\theta$ jointly. Each training step, the simulator provides trajectory $h = w^{\text{INIT}}a^0 \ldots a^{l-1}w^l$. $s_0^t$ $s_i^t$ represents the public state and information set of player $i$ at time $t$ in the same trajectory.

The training involves two additional losses, the first one $\mathcal{L}_\theta^A$ updates the $\Lambda_{i,\theta}$ and $\kappa_\theta$ using a soft clustering objective similar to fuzzy c-means (Bezdek et al., 1984). It minimizes a mean squared error between real and abstract properties weighted by the softmax. The second loss $\mathcal{L}_\theta^S$ updates the $\Lambda_{i,\theta}^I$ by minimizing the cross entropy loss between the predicted probability distribution over the abstract information sets and the one-hot encoded vector of the abstract information set that is the nearest neighbor of the real information set based on the $\kappa$.

$$\mathcal{L}_\theta^A = \sum_t^{l-1} \sum_{i \in \mathcal{N}} \sum_{\overline{s_i^t} \in \Lambda_{i,\theta}(s_0^t)} ||\kappa_\theta(\overline{s_i^t}) - \kappa(s_i^t)||^2 \frac{e^{-\gamma||\kappa_\theta(\overline{s_i^t}) - \kappa(s_i^t)||^2}}{\sum_{\overline{s_i^t}' \in \Lambda_{i,\theta}(s_0^t)} e^{-\gamma||\kappa_\theta(\overline{s_i^t}') - \kappa(s_i^t)||^2}} \tag{2}$$

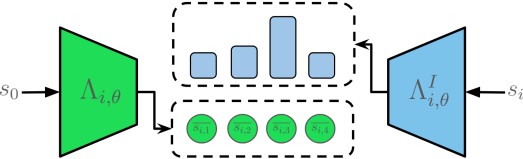

Figure 1: The public state and information set representations functions of player $i$. First the $\Lambda_{i,\theta}$ predicts 4 abstract information sets and then $\Lambda_{i,\theta}^I$ predicts the probability distribution over those abstractions.

$$\overline{s_i^{t,*}} = \arg\min_{\overline{s_i^t} \in \Lambda_{i,\theta}(s_0^t)} ||\kappa_\theta(\overline{s_i^t}) - \kappa(s_i^t)||^2 \qquad \mathcal{L}_\theta^S = \sum_t^{l-1} \sum_{i \in \mathcal{N}} \mathcal{L}_\theta^{CE}(\text{sg}(\overline{s_i^{t,*}}), \Lambda_{i,\theta}^I(s_i^t)) \qquad (3)$$

Here, $x = \text{sg}(x)$ is an identity operator that stops the gradient flow through $x$. The $\gamma$ controls the softness of the clustering and as $\gamma \to \infty$ the clustering becomes hard. The gradients from $\mathcal{L}_\theta^A$ propagate through both $\kappa_\theta$ and $\Lambda_{i,\theta}$, but the gradients from $\mathcal{L}_\theta^S$ are only propagated through $\Lambda_{i,\theta}^I$.

$$\mathcal{L}_\theta^{MA} = \mathcal{L}_\theta^M + \mathcal{L}_\theta^A + \mathcal{L}_\theta^S \qquad (4)$$

The overall loss also includes the model learning loss from Section 3. Importantly, $\mathcal{L}_\theta^M$ is computed using the dynamics based on the selected abstract information set. Furthermore, the gradients from $\mathcal{L}_\theta^M$ are not backpropagated through $\Lambda_{i,\theta}$, $\Lambda_{i,\theta}^I$ or $\kappa_\theta$. This decouples the learning of the abstraction structure from the learning of the model dynamics.

## 5 DEPTH-LIMITED SOLVING

While the learned model and abstraction allow reconstructing the whole game tree of the abstracted game, practical applications in large imperfect information games rely on depth-limited reasoning combined with a learned value function to estimate payoffs beyond the reasoning horizon. Defining and training optimal value function for imperfect information games is challenging, as they theoretically depend on belief states (Brown et al., 2020; Kovařík et al., 2023). The belief states are public states with corresponding probability distribution of reaching each history. Such value function is often trained by repeatedly sampling varied belief states and performing depth-limited reasoning in each of them Moravčík et al. (2017); Schmid et al. (2023), which is generally intractable.

In this section we present an algorithm *Learned Abstract Model for Imperfect-information Reasoning* (LAMIR), which learns the abstracted model from Section 4 along with a value function that enables depth-limit reasoning. We first show what additional components are necessary to learn the value function, then we summarize the whole training procedure and lastly we show how to use these trained components for depth-limited reasoning.

LAMIR uses a value function based on the multi-valued states (Brown et al., 2018; Kubíček et al., 2024; Milec et al., 2025). Training of this value function just from samples requires these additional components:

- Strategy function $\pi_\theta : \mathcal{S}_i \to \Delta\mathcal{A}$, which for given information set from the original game returns the strategy trained with some policy-gradient algorithm, like RNaD (Perolat et al., 2022).
- Transformations function $\tau_\theta : \mathcal{S}_i \to \mathbb{R}^{|\mathcal{A}_i| \times |T|}$, representing $T$ characteristic directions in strategy space explored by the policy-gradient algorithm during training. For a single transformation $\chi \in \tau_\theta$ the transformed strategy is computed locally as $\pi_i^\chi(s_i, a_i) = \pi_i(s_i, a_i) + \chi(s_i, a_i)$. The resulting strategy $\pi_i^\chi(s_i)$ is clipped to non-negative numbers and then normalized (Kubíček et al., 2024).
- Value function $v_\theta : \overline{\mathcal{S}_1} \times \overline{\mathcal{S}_2} \to \mathbb{R}^{|T| \times |T|}$, which approximates the expected value of each combination of transformed strategies between players (Kubíček et al., 2024).

We train $\pi_\theta$ and $\tau_\theta$ using real information sets from sampled trajectories. However, $v_\theta$ is trained using the corresponding joint abstract information sets instead of histories. It is possible that several

real histories map to the same abstract state with varying reach probabilities. This may introduce bias to the trained values. While importance sampling could correct this, we found in our experiments it did not affect the results in any significant way. This is most likely due to transformations being just a heuristic approach to approximate different parts of the strategy space.

We have used Regularized Nash Dynamics (RNaD) as the policy-gradient algorithm, which includes Neural Replicator Dynamics (NeuRD) loss for strategy training and mean squared error for the associated baseline value function $\mathcal{L}_\theta^{PG}$ (Hennes et al., 2020; Perolat et al., 2021; 2022). Note, that the value function from the RNaD cannot be used as a value function for look-ahead reasoning, because it represents value for a specific belief based on the networks strategy and not for a arbitrary belief. We use mean squared error for the training of the value function $\mathcal{L}_\theta^V$ and the targets are computed by the V-trace (Espeholt et al., 2018; Kubíček et al., 2024), which estimates the value of different policies in an off-policy setting.

Following Kubíček et al. (2024), transformations represent the strategy changes during training. For each player $i$ we compute the difference vector $\delta_i^t = \pi_i^{t,\text{NEW}} - \pi_i^{t,\text{OLD}}$, where $\pi_i^{t,\text{OLD}}$ and $\pi_i^{t,\text{NEW}}$ are concatenated strategies along the whole trajectory before and after the training step. Instead of the hard clustering proposed originally, we use the soft clustering from Section 4.

$$\mathcal{L}_\theta^T = \sum_{i \in \mathcal{N}} \sum_{\chi_i \in T} ||\chi_i - \delta_i^t||^2 \frac{e^{-\gamma ||\chi_i - \delta_i^t||^2}}{\sum_{\chi_i' \in T} e^{-\gamma ||\chi_i' - \delta_i^t||^2}} \tag{5}$$

## 5.1 TRAINING

A single training iteration of LAMIR begins by sampling a trajectory $h = w^{\text{INIT}} a^0 \ldots a^{l-1} w^l$ according to a sampling strategy $\mu$. In the on-policy setting, this strategy is the current policy, $\mu = \pi_\theta$. While our approach is not limited to on-policy learning, off-policy settings would require scaling the policy and value function updates with importance sampling.

For any sub-history $h^t \sqsubseteq h$, the simulator provides the true public state representation $s_0^t(h^t)$, true information set representation $s_i^t(h^t)$, legal actions $\mathcal{A}_i^t(s_i^t)$, and the reward $r^t$. Our method uses a two-step update per episode. This is necessary because the transformation loss $\mathcal{L}_\theta^T$ depends on strategy changes induced by the policy-gradient step. First step trains the strategy function $\pi_\theta$ using a policy-gradient loss $\mathcal{L}_\theta^{PG}$ of the underlying policy-gradient algorithm. In the specific case of RNaD, this loss is NeuRD (Hennes et al., 2020). Second step computes the transformation loss $\mathcal{L}_\theta^T$ and value function loss $\mathcal{L}_\theta^V$ (Kubíček et al., 2024). The second step also includes all losses related to learning the game abstraction.

At each timestep $t$, the abstraction network $\Lambda_{i,\theta}$ predicts a set of $L$ possible abstract information sets for each player: $\overline{s_{i,j}^t} = \Lambda_{i,\theta}(s_0^t)$. The goal is to cluster information sets in a $K$-dimensional space. A trainable function $\kappa_\theta(\overline{s_{i,j}^t})$ maps the abstract information sets into this space. Similarly $\kappa(s_i^t)$ maps the true information set. $\kappa$ could be a separate provided function either by environment or user, but similarly it could be the trained strategy $\kappa = \pi_\theta$. $\mathcal{L}_\theta^A$ is computed using the $K$-dimensional representations. It updates both $\Lambda_{i,\theta}$ and $\kappa_\theta$.

Concurrently, a separate network $\Lambda_{i,\theta}^I$ predicts a probability distribution over the $L$ possible abstract sets $\overline{s_{i,j}^t}$ given the current real information set $s_i^t$. The $\mathcal{L}_\theta^S$ is a cross-entropy loss that trains $\Lambda_{i,\theta}^I$ to increase the likelihood of selecting the closest abstraction, $\overline{s_i^{t,*}}$ in the $K$-dimensional space. Finally, the $\mathcal{L}_\theta^M$ is computed as described in Section 3. The dynamics model $\Upsilon_\theta$ is trained to predict the most likely future abstraction, $\overline{s_{i,j}^{t+k,*}}$, after $k$ steps. The final losses for the two-step update are

$$\mathcal{L}_\theta^1 = \mathcal{L}_\theta^{PG} \qquad \mathcal{L}_\theta^2 = \mathcal{L}_\theta^{MA} + \mathcal{L}_\theta^V + \mathcal{L}_\theta^T \tag{6}$$

## 5.2 LOOK-AHEAD REASONING

At test time, LAMIR employs continual resolving (Moravčík et al., 2017; Schmid et al., 2023) within the learned abstract game. The process begins at the initial history $h^{\text{INIT}}$, which defines

the starting public state $s_0(h^{\text{INIT}})$ and information sets $s_1(h^{\text{INIT}}), s_2(h^{\text{INIT}})$. The root of the initial abstract subgame is generated using $\Lambda_{i,\theta}, \Lambda_{i,\theta}^I$.

The following process then repeats until the terminal state is reached. The algorithm constructs the depth-limited game tree using the learned dynamics $\Upsilon_\theta$ and legal actions $\Gamma_\theta$. At the depth limit, a final "decision layer" is added for non-terminal states. This layer consists of $T$ artificial actions where both players choose their strategy for the remainder of the game. Each combination leads to a terminal state with a reward defined by the learned value function $v_\theta$ (Brown et al., 2018; Kubíček et al., 2024).

In this newly created abstracted game LAMIR runs Counterfactual Regret Minimization+ (CFR+) or some of it's variant (Zinkevich et al., 2007; Tammelin, 2014; Farina et al., 2019) to get a strategy at each decision node. The actual current information set of the acting player $s_i$ is mapped to its corresponding abstract information set $\overline{s_i}$ using $\Lambda_{i,\theta}^I$. The strategy corresponding to $\overline{s_i}$ is used to sample an action that advances the real game to the new state.

Out of the previous game tree, all the histories consistent with $s_0$ are used to create a new gadget game. Each of those histories is associated with joint abstract information set. Thanks to $\Lambda_{i,\theta}$, there are at most $L^2$ unique combinations. Any two histories that share the same joint abstract information set are merged so that the new subgame root contains at most $L^2$ nodes. Counterfactual values and the acting player's reaches are reused from the previous subtree to provide necessary statistics to the gadget game. The algorithm repeats this process until it reaches the terminal state (Burch et al., 2014; Moravčík et al., 2017).

# 6 EXPERIMENTS

## 6.1 EXPLOITABILITY IN SMALLER GAMES

To ensure that the strategies found by LAMIR approximate Nash equilibria, we applied it in games small enough to compute exact exploitability, which serves as a distance from the Nash equilibrium in two-player zero-sum games. For various abstraction sizes $L$ and different properties for clustering $\kappa$ we trained LAMIR with 10 different random seeds for 100,000 episodes. In all of our experiments we used Regularized Nash Dynamics (RNaD) to train the strategy for multi-valued states value function. The rules of the games used in experiments are in Appendix C.

Starting at episode 80,000 we have computed exploitability every 1000 episodes. In each public state, the algorithm uses the trained functions to construct the depth-limited subgame with depth limit 1. Then this subgame is solved using CFR+ and the strategy from abstract infosets is mapped to real ones. Then we compute the exploitability of the final composed strategy in the original game. The results for different $\kappa$ and $L$ are displayed in Figure 2.

Compared to RNaD, the LAMIR has a more profound training. As a result single training step of LAMIR takes more time. In our implementation, the training is roughly 2-2.5 times slower per-iteration depending on the size of the abstraction $L$. We further discuss this in Appendix B.2. Still, the exploitability of RNaD reported in Figure 2 does not improve even with increased training time. The main likely cause are network approximation errors. Similarly in the original results, RNaD could not decrease the exploitability in Leduc Poker beyond a certain point (Perolat et al., 2021).

We used three different $\kappa$, which served as a basis for clustering, first the legal actions, second the legal actions with the current RNaD strategy, and third the legal actions, RNaD strategy, and the player's action history. Each choice of $\kappa$ is capable of outperforming the concurrently trained RNaD with the same amount of training episodes given sufficient $L$. When using the action history as $\kappa$, each information set is uniquely defined in a given public state, which means that, with sufficient capacity, the dynamics network should model the underlying game. This was evaluated in II Goofspiel 5 with $L = 30$, which indeed has mapping one-to-one for each abstract and information set. The main reason why the exploitability is greater than 0 arises from the discrepancies in the rewards produced by the dynamics function.

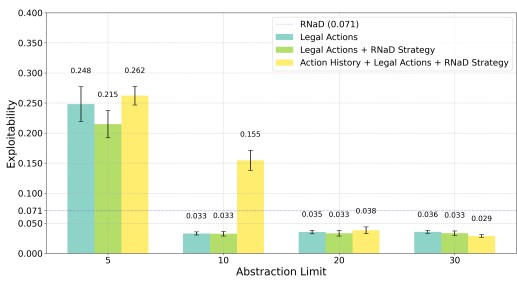 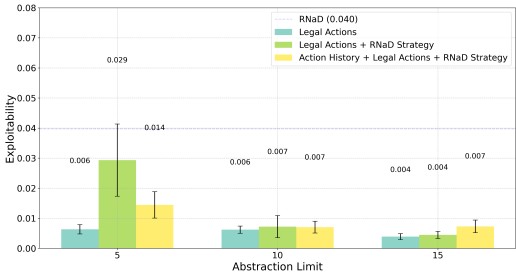

(a) Imperfect Information Goofspiel 5  (b) Imperfect Information Oshi-Zumo 3,5

Figure 2: Exploitability of LAMIR in a different games by using continual resolving with depth-limit 1 in each subgame with different choice of abstraction limit $L$ and $\kappa$. The largest public state in II Goofspiel 5 contains 30 infosets and in Oshi-Zumo 3,5 it contains 625 information sets.

| Algorithm | II Goofspiel 10 | II Goofspiel 13 | II Goofspiel 15 |
|---|---|---|---|
| LAMIR $\kappa$ = Legal actions | $54.47 \pm 0.25$ % | $60.68 \pm 0.34$ % | $80.49 \pm 0.26$ % |
| LAMIR $\kappa$ = RNaD strategy | $61.60 \pm 0.29$ % | $58.33 \pm 0.27$ % | $61.80 \pm 0.36$ % |

Table 1: Average win rate with 2-sigma error bars of LAMIR against RNaD in different games.

## 6.2 HEAD-TO-HEAD IN LARGE GAMES

The main usage of LAMIR is in very large games, where the exact exploitability cannot be computed. This experiment evaluates LAMIR in this exact setting, where we compare it with Regularized Nash Dynamics (RNaD) in head-to-head play. For each game and $\kappa$, we have trained LAMIR with 3 different random seeds for 3 million episodes. Similarly, we have trained RNaD with 6 different random seeds for the same number of episodes. The hyperparameter settings remained the same for each game. Then we matched each trained seed of LAMIR with each trained seed of RNaD and played more than 100,000 matches. Note that when we use $\kappa$ = RNaD strategy, it is the strategy that is learned concurrently for the value function and not the RNaD strategy against which the algorithm is compared later. The resulting win rates with 2-sigma error bars are in Table 1.

LAMIR consistently outperforms RNaD in each of the tested games. Imperfect Information Goofspiel is known for its large public states, so continual resolving without abstractions is not applicable. SePoT (Kubíček et al., 2024) was also evaluated in such large domains, but we did not compare against it, as it only uses CFR if the subgame is small enough. The authors showed that in II Goofspiel 13, SePoT has a win rate of only 52%, which is likely caused by not resolving almost any subgame due to the limit on the size of the subgame. Furthermore, we show that even in larger games than those tested with SePoT, LAMIR improves over RNaD even more.

## 7 LIMITATIONS AND FUTURE WORK

LAMIR advances the scalability of the continual resolving paradigm to larger games by integrating learned models and abstractions. However, it presents several interesting avenues for future research.

The computational complexity of the look-ahead reasoning, when using CFR is linear in the amount of information sets in the game. This complexity still remains, as LAMIR only reduces the size of the game in each public state but uses CFR in the abstract game. Each subgame LAMIR construct with depth $D$ contains at most $\sum_{d \in \{0,...,D\}} L^2 |\mathcal{A}|^{2d}$ unique nodes and at most $L^2 |\mathcal{A}|^{2D} T^2$ terminal histories, where $T$ is the amount of transformations. Most of the subgames would be smaller, but this is the main limitation of LAMIR as it limits how large $L$ can be.

Currently, LAMIR's dynamics network $\Upsilon_\theta$ does not explicitly model chance nodes within the game. LAMIR could still be applied in games with chance events, but this is not an intended setting as the absence of chance nodes will result in poor abstraction, regardless of the abstraction capacity $L$. Although algorithms like Stochastic MuZero (Antonoglou et al., 2022) demonstrate that modeling chance is feasible in learned models for perfect information games, integrating chance nodes into

our framework, particularly in conjunction with learned abstractions, requires careful consideration and is a key area for future work.

The effectiveness of the learned abstraction depends on the chosen property function $\kappa$ for clustering. In games without chance, if this property function could perfectly distinguish two different information sets, $L$ is greater than the size of the largest public state and neural networks have sufficient capacity, then LAMIR learns a near-perfect model of the game. When $L$ is reduced to achieve scalability, the learned abstraction may not capture all crucial strategic nuances, which can affect the strength of the derived strategy. In experiments, we have used simple proxies present in any game and it still yielded strong performance. Ideally, $\kappa$ would also be learned during the training process.

Reducing $L$ necessarily introduces imperfect recall, meaning that the player in the abstract game may "forget" information it previously knew. Algorithms like Counterfactual Regret Minimization (CFR) guarantee convergence to a Nash equilibrium strategy only in perfect recall games. While CFR's convergence is not generally guaranteed in imperfect recall settings, it has been shown for subclasses such as A-loss recall games (Čermák et al., 2020). Our abstractions are A-loss recall games if the public observations in the original game depend only on prior public information and the actions taken in the current round. Games like Imperfect Information Goofspiel or Oshi-Zumo satisfy these conditions, but many others, including Battleships, Dark Chess, or Stratego, do not. Thus, for games that do not fall into the A-loss recall category after abstraction, theoretical convergence guarantees for CFR-based methods within LAMIR are not assured.

LAMIR focuses on abstracting information sets but does not inherently abstract action spaces. In games with very large or continuous action spaces (e.g., bet sizing in Poker), the sheer number of actions can remain a bottleneck for the look-ahead reasoning, regardless of the information set abstraction. While action abstractions have been extensively studied (Brown & Sandholm, 2014; 2018; Li et al., 2024), integrating them with LAMIR is out of the scope of single paper, but it presents another direction for future improvement.

# 8    CONCLUSION

In this paper, we have introduced LAMIR, an algorithm designed to enable look-ahead reasoning by learning the model of the game with a suitable abstraction directly from experience, without the need for any domain-specific knowledge.

Our core contributions are fourfold. First, we have proposed a method for learning the fundamental components of a model of dynamics of imperfect information games without chance. Second, we developed a technique for automatically learning an abstraction by clustering information sets, effectively reducing the size of the game. Third, we integrated these components with a learning of value function in the abstracted game that enables depth-limited look-ahead reasoning. Lastly, we demonstrate how LAMIR facilitates the continual resolving paradigm by performing a depth-limited look-ahead reasoning in each decision node encountered.

We empirically verify that, when given sufficient capacity, LAMIR learns a nearly perfect model. Still, the game-playing capabilities degrade gracefully when the abstraction capacity is reduced. We have also shown that LAMIR manages to perform look-ahead reasoning even in intractably large public states. Thanks to that, it achieved up to 80% win rate in large games compared to RNaD, a strong baseline that was successfully used to create a human-expert level player in Stratego.

The primary impact of LAMIR lies in scaling look-ahead reasoning techniques to larger games by learning abstraction and its model directly from experience. This overcomes the most notable limitations of the continual resolving paradigm, which requires each subgame to be tractable and to have explicit access to the game model. LAMIR has several limitations, which can be addressed in future work. Still, LAMIR is, to the best of our knowledge, the first algorithm that enables look-ahead reasoning in large-scale games like Imperfect Information Goofspiel 15 without any domain-specific knowledge, substantially outperforming the model-free policy-gradient algorithm.

## ACKNOWLEDGMENTS

This research is supported by Czech Science Foundation (GA25-18353S) and Grant Agency of the CTU in Prague (SGS23/184/OHK3/3T/13). Computational resources were supplied by (e-INFRA CZ LM2018140) supported by the Ministry of Education, Youth and Sports of the Czech Republic and also (CZ.02.1.01/0.0/0.0/16 019/0000765)

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
