# OpenReview forum: "Look-ahead Reasoning with a Learned Model in Imperfect Information Games"
_ICLR.cc/2026/Conference — ICLR 2026 Poster_

### Official Review · Reviewer_MHic · 2025-10-23

**Soundness:** 3
**Presentation:** 3
**Contribution:** 3
**Rating:** 6
**Confidence:** 3

**Summary:**

The authors proposed a novel algorithm that enables subgame search in imperfect-information games with a large number of public states. Specifically, they trained a clustering network to group similar information sets and solved the resulting abstracted subgames using CFR+. In addition, to constrain the size of the subgame, they trained a value network to predict state values, enabling depth-limited subgame search.

**Strengths:**

- The performance of the algorithm looks good. It outperforms RNaD consistently.
- The writing of the paper is clear.
- The paper enables subgame search in games with a large public states.

**Weaknesses:**

- The techniques presented in this paper are not particularly novel. It appears to combine several existing approaches -- subgame search, game modeling, and abstraction. That said, integrating all of these components simultaneously is a non-trivial task.
- The games evaluated in this paper do not seem to require model learning, as used in MuZero. I would prefer to see the method tested on games where developing an accurate simulator is intractable.

**Questions:**

See Weaknesses.

---

> ### Author Response · Authors · 2025-11-19
>
> We would like to thank the reviewer for their review.
>
> Weakness 1: We agree that we build upon the other notable works in the field.
>
> Weakness 2: “Games in experiments.” The games we have tested have simple rules, but they are large in size. Goofspiel 15 can have public states (i.e. root of subgames) greater than 10^18. In such large subgames, it is not possible to do sound test-time reasoning with conventional methods. It is possible to create hand-crafted abstractions of such games that would keep those subgames tractable. However, the goal of LAMIR is to find these abstractions purely in self-play without relying on human input, which could be laborious and prone to error. We agree with the reviewer that testing on games with more complicated models, such as video games, would strengthen the paper's claims, but we believe this is outside the scope of a single paper.

---

> > ### Comment · Reviewer_MHic · 2025-11-24
> >
> > Thank you for your reply, and I will maintain the rating.

---

### Official Review · Reviewer_q3UX · 2025-10-28

**Soundness:** 3
**Presentation:** 2
**Contribution:** 3
**Rating:** 6
**Confidence:** 4

**Summary:**

This paper introduces LAMIR (Learned Abstract Model for Imperfect-information Reasoning), an algorithm designed to enable look-ahead reasoning in large-scale imperfect information games by an abstract model, which can directly learn from experience without any domain-specific knowledge. LAMIR achieves up to an 80% win rate in large imperfect information games compared to RNaD, and also shows robustness in some smaller games.

**Strengths:**

1. This paper provides a novel approach that combines model learning, abstraction techniques, and value function learning to address scalability challenges in large-scale imperfect-information games.

2. The comparison to the baseline, RNaD, is persuasive, as it was previously used in DeepNash in Stratego.

3. The experimental figures and methodology descriptions are clear and well-structured, providing a comprehensive understanding.

4. The future work proposed by the authors is sound. For instance, integrating abstract action spaces could further enhance the applicability of LAMIR to various types of games, particularly those with large or continuous action spaces.

**Weaknesses:**

1. The description in Section 5 is insufficient. In particular, the explanations of the training and testing processes should be clarified, as they are somewhat difficult to follow in the current version. This is especially important because Section 5 presents the overall algorithm that integrates all components discussed in the previous sections. Including a flowchart or pseudocode would help readers better understand the steps.

2. Section 4 uses a significant amount of space to describe the clustering process and explains how to train the clustering of the public state. However, it seems that this approach is applied only to the root node of the test-time search tree. The tree search process requires a more detailed description, including how each component introduced in previous sections is used.

3. Since LAMIR requires clustering during training, the overall training time is expected to increase. The paper should clearly report the additional training cost of LAMIR and compare it with previous approaches, such as RNaD. It would also be helpful to add an additional experiment discussing how the computational cost increases as the number of clusters $L$ increases, as well as how much data is required to train both the clustering and abstraction processes.

4. As mentioned in Section 7, the proposed approach seems to be constrained by the limitations of the CFR algorithm, where it is not applicable to many other games like Dark Chess and Stratego. However, the abstraction method itself should not be affected by this. Have there been any attempts to apply the core ideas of abstraction to these games?

5. The experiments are conducted on relatively simple imperfect information games. Are there any experiments on more complex games, such as poker? Including such results would make the work more convincing by better demonstrating the generality and effectiveness of LAMIR.

**Questions:**

Please address each concern raised in the weaknesses.

### Regarding the training process:
1. How is $\Lambda_{i,\theta}$ (the green component in Figure 1) specifically trained? Is it implemented using a clustering algorithm, or is it also modeled as a neural network? If it is a neural network, what are its input and output representations? Does it use the clustering result as the training target? It seems that the training loss in equations (2) and (3) is related only to $\Lambda^{I}_{i,\theta}$.

2. In equation (2), during training, does it require enumerating all $\overline{s^t_i}$? Or is it simply using sampling? If it is sampling, how many samples are required? Is this sampling process required to be performed for every public state? What is the computational cost of this sampling process? If the number of samples is too small, is the clustering into $L$ groups still effective? How is the value of $L$ determined in practice? Does the number of samples increase as $L$ increases? If so, how does the computational cost scale with $L$? Overall, what is the additional computational cost of training LAMIR compared to RNaD?

3. What kind of training data is used? Is it based on a pre-defined dataset or trajectories obtained from its self-play?

### Regarding test time:
1. Section 5 presents the overall algorithm that integrates all components discussed in the previous sections. However, the description in Section 5 is unclear, particularly regarding how the components are used in the tree search. It would be helpful to include a figure illustrating the test-time architecture, along with a more detailed explanation of each component.

2. During the search tree, is it still required to enumerate all $\overline{s^t_i}$ or sample $\overline{s^t_i}$ to obtain the latent abstract information sets?

3. It seems that $\Lambda_{i,\theta}^{I}$ is used only at the root node of the game tree, where a specific abstraction set is chosen. However, this abstraction set only represents a single category. After performing a sequence of actions during the search tree, can we guarantee that this category still adequately covers all possible scenarios at the leaf nodes? Or is $\Lambda_{i,\theta}^{I}$ repeatedly used during the search at each internal node?

4. The inference time is not reported. Currently, the depth is set to 1. Why not use a larger depth? It would be helpful to include an ablation study analyzing different depth settings, along with a discussion of the additional computational cost and potential performance gains as the depth increases.

### Regarding the experiments:

1. How are the different bases for clustering ($\kappa$) implemented?
2. Regarding training and test time, are LAMIR with different $L$ values and RNaD all trained under the same training time limits?
3 . Based on the results in Table 1, why does using Legal Actions for clustering lead to a higher win rate than the RNaD strategy in the Goofspiel 13 and Goofspiel 15 environments? Additionally, would it be fair to use the RNaD strategy for clustering within LAMIR and then compare it directly with RNaD itself, given that LAMIR would already know the RNaD strategy?

### Other questions and suggestions
1. The reference formatting in the main text is incorrect. Most citations are missing brackets.
2. How are the public state and the player's information set generated during training, especially in large-scale games? Additionally, what is the approximate number of public states in the "large games" used in the experiments?

---

> ### Author Response · Authors · 2025-11-19
> **Response to weaknesses**
>
> We would like to thank the reviewer for their extensive review. We appreciate the non-trivial time this must have taken.
>
> Weakness 1:  Thank you for pointing out the details that were not clear from the paper. We have expanded Section 5 and included pseudocode and a figure explaining the continual resolving within LAMIR in Appendix D. These changes should clarify the role of individual components both during training and at test-time reasoning. Please let us know if this improved the explanation.
>
> Weakness 2: It is true that in the resolving part, the $\Lambda_{i, \theta}$ is used to construct the root of the subgame at test time. However, the clustering is used at each public state during training, not only the initial one (root of the game), and it is crucial to the algorithm, as it keeps all the subgames tractable. We have revised the second part of Section 5, so that it explicitly states how the $\Lambda_{i, \theta}$ keeps all the subgames tractable.
>
> Weakness 3: Thank you for pointing this out. Indeed, the training time increases per iteration. LAMIR will train at least 8 networks (Public state abstraction $\Lambda_{\theta}$, infoset abstraction $\Lambda_{\theta}^I$, function $\kappa_{\theta}, dynamics function $\Upsilon_{\theta}$, legal actions  $\Gamma_{\theta}, Strategy  $\pi_{\theta}$, multi-valued states $v_{\theta}$, transformation function $\tau_{\theta}$, compared to 2 networks (strategy and value function) used by policy-gradient algorithms like RNaD. So the LAMIR could be roughly 4-times slower per iteration. We have trained $\Lambda_{\theta}, \Lambda_{\theta}, \Gamma_{\theta}$ for each player separately, which results in 11 networks. In our implementation, LAMIR training is roughly 2-2.5 times slower than RNaD per iteration, depending on the size of the abstraction $L$. The reason is that the part of the training is trajectory sampling and computing the V-Trace loss is more computationally expensive
>
> However, in the small games used in Section 6.1, RNaD cannot further improve beyond the reported values even with an increased amount of samples due to errors in the function approximators. Similar behavior was also observed in the original paper (Perolat et. al. 2021 Figure 3). So in those games, even with the same training time, LAMIR would still outperform RNaD. We have added a note about this in our revision to section 6.1.
>
> LAMIR does not require more traces or a different sampling scheme than the standard policy-gradient algorithms, i.e., in Section 6, we have used the exact same amount of traces, sampled in the same way (but with different seeds) for RNaD and LAMIR. The clustering we use is just an approximate online version, not the tabular clustering. It is highly likely that tabular clustering would produce better results. This is an ablation study we have tried in Appendix B.1.
>
> Regarding the number of samples required as $L$ increases, this depends on the size of the public state and the chosen property function $\kappa$, and it directly corresponds to the complexity of online clustering. However, the goal of our approach is to leverage the generalization capabilities of neural networks to “share” the knowledge about clustering between similar public states.
>
> Weakness 4: We are not sure we understand the question correctly, so please clarify if we do not answer it satisfactorily here. We agree that the abstract model is not affected by the constraints of the CFR in the original games, so it is possible to apply it in games like Stratego or Dark Chess. We have not applied it directly to those games, but we have used Goofspiel 15, where the usual CFR cannot be used without abstraction (more on that in response to Weakness 5). The note we had in the Section 7 about this limitation was that the CFR limitation puts a constraint on the size of the abstraction $L$, as using $L > 10^6$ would make root of the subgame as large as $L^2 = 10^{12}$ and in games like Dark Chess where player can have ~50 moves, such subgame contain almost $10^{14}$ states which is already generally intractable. So LAMIR can be applied to Dark Chess, but it would require setting $L$ to a reasonable amount, such that the subgames remain tractable.
>
> Weakness 5: The games we have tested the LAMIR on are simple, but large. The Imperfect Information Goofspiel 15, which we have used in experiments, contains public states with more than $10^{18}$ unique states, which is much larger than in Poker. Even with the simplistic rules of the game, the public states grow quickly because the public information consists only of a sequence of Wins/Losses/Draws, and the cards played by both players define each unique state. There may be a hand-crafted abstraction that reduces the public state size, but the goal of LAMIR is to find these abstractions from the training data without any prior domain knowledge. In the revision, we have added the table to the Appendix, which illustrates the sizes of the used games.

---

> ### Author Response · Authors · 2025-11-19
> **Response to training and test time**
>
> Regarding the training process:
>
> 1. In the experiments we have used $\Lambda_{i, \theta}: \mathbb{R}^{|s_{0}|} \to \mathbb{R}^{L \times S}$, where $\theta$ are learnable parameters of the neural network, $L$ is the amount of abstractions and $S$ is the size of the latent representation of a single abstraction ($L$ and $S$ are hyperparameters of the algorithm). So each abstract infoset is represented in a latent $S$-dimensional space. Note that this is not the only approach. In Appendix Table 3, we provide $S$ for each experiment. A different approach could be similar to DreamerV2/V3, which produces a categorical distribution over learned features. We have decided not to put constraints on what the output shape of the abstraction function should be, so that the framework is more general. The equation (2) is then used directly as a loss function to update the weights of the neural network $\theta$. Regarding the second note about equations (2) and (3), we have changed equation (3) so that it is clearer what the role of $\Lambda_{i, \theta}^{I}$ is in the loss.
>
> 2. The entire training process, including equation (2), utilizes the exact same sampled traces as vanilla RNaD, with no additional data.  So, it does not require any enumeration during test time, which is intentional, as our setting assumes only a simulator of the game, not necessarily the model of the game. This is a usual reinforcement learning setting that is present in many scenarios, e.g., operating robots, playing video-games, etc. As a result, we cannot assume that we will have a sufficient number of samples in each public state. It is true that, in some public states, the clustering may not have enough samples, but the goal is to leverage the generalization capabilities of the neural networks to transfer knowledge between similar public states. It is correctly pointed out that with increasing $L$, more samples are required to train good cluster centers, but in the training, we do not assume that some public states would be visited more due to increased $L$. As a result, the choice of hyperparameter $L$ depends on the size of the game and the expected amount of sampled trajectories, e.g. more samples allow for greater $L$, increased game size will introduce more points where to cluster, so it may be beneficial to choose smaller $L$ if the amount of samples remains the same, but the game is larger.
>
> 3.The game trajectories are sampled purely through self-play with a simulator. The training process can only submit actions of all players and see their observations, but it cannot manipulate the internal state of the simulator, i.e., it cannot start from an arbitrary point in the game.
>
> Regarding test time:
>
> 1. We have addressed this in Weakness 1.
>
> 2. No,  enumerating the real information set $s_{i}^t$ at each public state is not required, neither in training nor in testing. We assume it is not feasible to do it at all. In testing, LAMIR generates the abstract information sets with $\Upsilon_{\theta}$ and $\Lambda_{i, \theta}$, which requires only the public state representation $s_0$, which is part of the information set representation $s_i$ that is provided to the player by the simulator.
>
> 3. Indeed, $\Lambda_{i, \theta}$ and $\Lambda_{i, \theta}^I$ are used only in the root to generate the initial subgame, while the dynamics function $\Upsilon_\theta$ is used to generate the rest of the subgame tree. Then, after performing an action and receiving observation (which induces a new information set and public state), the consistent part from the previous subgame is used as the root of the new subgame, which is again generated by $\Upsilon_\theta$. It is correctly pointed out that these new subgames may gradually become inconsistent with the $\Lambda_{i, \theta}$. It is possible to enforce this consistency by further means. However, with sufficient training time, this consistency should naturally emerge, and in our experiments, we have not encountered this to be an issue, so we decided not to include it in the method as it simplifies the description.
>
> 4. Thank you for pointing this out. We have added experiments to the Appendix that demonstrate how increasing the depth-limit affects performance. Note that increasing the depth-limit does not require retraining the networks, only more resources at test time. For this experiment, we have used the trained networks from the main body of the text. These additional results show that when the abstraction size $L$ is small, increasing the depth limit degrades performance. This happens as the poor abstraction can be partially fixed by the value function. However, as the $L$ increases, the exploitability is either on par with depth-limit 1 or slightly better.

---

> ### Author Response · Authors · 2025-11-19
> **Response to experiments and other questions**
>
> Regarding the experiments:
>
> 1. The bases of $\kappa$ are as follows: Legal actions are represented by a binary vector of size $ | \mathcal{A}_i | $ (The number of actions in the game). For example, in Goofspiel with N cards, this vector has size exactly N, and if the card with value 6 is a legal action, then the binary vector has at position 6 value 1 (otherwise it would be 0). The strategy is a vector of the same size, but the values are now continuous between 0 and 1. Action history is a binary matrix of size $|\mathcal{A}_i| \times T$, where $T$ is the length of the game, where 1 at (i, j) represents that action with index $i$ was played at turn $j$. This matrix is then flattened into a vector. A combination of the different properties is used by a simple concatenation of these individual components.
>
> 2. The RNaD and LAMIR are not trained with the same time-limits, but with the same amount of available samples. Because LAMIR trains more networks, it takes more time (roughly 2-2.5 times in our experiments), but it achieves better sample complexity and better convergence. It is important to note that we have used games with very fast simulators. One of the LAMIR strengths is that it only requires sampled traces from the simulator, so it can be applied even in settings where the simulator itself may be the limiting factor compared to the training time.
>
> 3. We believe that the superior performance of Legal actions compared to the RNaD strategy, as $\kappa$, is twofold. First, the strategy is not static during training, so clustering with respect to this ever-moving property is more difficult than clustering with respect to the legal actions, which remain static throughout the whole training. Second, we have used a small abstraction size $L$ compared to the size of the game, and the legal actions could be a better predictor of which information sets to cluster in Goofspiel. We believe that using the same RNaD strategy for the clustering within LAMIR and then comparing against it would not be fair. That is the reason why the RNAD, which is trained within LAMIR and the RNaD for testing, were trained completely separately on different trajectories with different initializations, so that we would minimize the risk of LAMIR abstraction being influenced by the strategy it is compared against. We discuss this at the end of the first paragraph in Section 6.2.
>
> Other questions and suggestions
>
> 1. Thank you for pointing this out. We have fixed this in the revision.
>
> 2. When we talk about information sets and public states during training, we do not mean all states consistent with the public state/information set. Instead, we mean the representation of information that the public state/information set contains. So, for example, in Stratego, the information set representation would be: The board, where the player knows the rank of all of its pieces, but only knows opponents' ranks for pieces that were in some duel. The public state representation would then contain only the ranks of pieces that were in a duel. As a result, we have used information sets/public states in training as a compact representation of the information provided by the environment itself. We did not use any additional enumeration. We have added the sizes of the games we have used to the Appendix.

---

> > ### Comment · Reviewer_q3UX · 2025-11-27
> >
> > Thank you for the detailed responses with the figure and algorithm in the appendix, which help a lot in understanding this work. I still have some problems, listed below.
> >
> > ###  Regarding the added Figure 7:
> >
> > 1. In Figure 7, if I understand correctly, this is for the inference. I am still confused about the whole training process. Do you have the figures for training? Especially, in the fifth paragraph of Section 5, I don’t understand the difference between the value function used in $\mathcal{L}^{PG}$ and $\mathcal{L}^{V}$. How to calculate these two loss functions?
> >
> > 2. In the inference part, apart from the first step of the whole game, how do other steps build a search tree? For example, in the right part of Figure 7, do you expand all the abstract nodes for the leaf? In the left part of Figure 7, do the root nodes of the search tree inherit from the previous search tree (all leaf nodes from the last step)? If we expand all the abstract nodes for the leaf and the root nodes inherit from the previous search tree, won’t the size of the search tree increase exponentially during the game progress? If not all abstract nodes are expanded and the root nodes inherit from the previous search tree, won’t some information disappear in the root nodes? In addition, if the root nodes inherit from the previous search tree, the clustering process will be only used in the initial state of each game, which does not seem useful.
> >
> > 3. In the right part of Figure 7, if all possible combinations of two players’ abstractions are generated, why should we learn the abstraction distribution?
> >
> > 4. According to the description in Figure 7, line 1215 mentioned that "the player samples an action $a^0_1$ from the strategy of the most likely abstraction”, but line 323 mentioned that "We train $\pi_\theta$ and $\tau_\theta$ using real information sets". Is the input of the strategy function a real information set $s_i$ or an abstract information set $\bar{s_i}$?
> >
> > ### Regarding the fairness comparisons/computational cost in the experiment:
> >
> > 1. Currently, LAMIR is roughly 2-2.5 times slower than RNaD, which is not really a fair comparison. Can you provide additional experiments comparing LAMIR and RNaD under the same training time?
> >
> > 2. Thanks for the revision in Section 6.1. Can you provide a table for the computational cost of different $L$?
> >
> > ###  Regarding merging abstraction:
> >
> > 1. In the right part of Figure 7, it seems the two abstract states merge into one state after the transition function. How to conduct the merge process?
> >
> > 2. In our previous reviews, "After performing a sequence of actions during the search tree, can we guarantee that this category still adequately covers all possible scenarios at the leaf nodes?", the author replied that this issue doesn’t appear in the experiments and said "with sufficient training time, this consistency should naturally emerge". Do you have any evidence that this will happen? Is there any theoretical proof to guarantee this? I am still concerned that the abstractions cannot cover all possible scenarios when the search goes deeper. For example, in the additional experiments in Figure 5, it seems that using deeper search results in worse performance. This might be caused by the merging abstraction problem. Do you have any experiments to verify this?
> >
> > ###  Minor problem:
> >
> > 1. The caption in Table 4 is missing.

---

> > > ### Author Response · Authors · 2025-12-03
> > >
> > > Thank you for further engaging in the review and for providing more notes on how should we improve the paper.
> > >
> > > ## Regarding the added figure 7.
> > > 1. Yes, you correctly pointed out that this is only inference. We intentionally did not specify $\mathcal{L}^{PG}$ as our approach is compatible with any policy-gradient approach and different approaches train different value functions to speed up the learning process of the policy. Similarly we did not specify $\mathcal{L}^{V}$, which was already proposed in prior work [1] and we discussed only the changes required for the abstract model. In our experiments, we have used V-trace to get the target values for both, but key difference is: $\mathcal{L}^{PG}$ trains single value for each information set, $\mathcal{L}^V$ trains several values for each joint abstract information set.
> > > 2. The root of the new search tree is always inherited from the previous subgame resolve. Note that this does not have to be from the leaves, if the depth-limit is greater than 1. Also you have correctly pointed out that if we would just naively take this as a root and construct a new depth-limited tree out of all those states, then we will not avoid the exponential blowup. This is connected to your other questions about merging the abstraction. The abstraction networks $\Lambda_{i, \theta}$ are trained so that each public state is abstracted to at most $L$ abstract information sets of each player as a result each abstract public state consists of at most $L^2$ states, because states in abstracted game are defined as joint abstract information sets. This means that even if we select more than $L^2$ nodes from the previous subgame as our new root, then each of those nodes need to correspond to one of the $L^2$ states. With that knowledge it suffices to map each node to one of those $L^2$ states. In the experiments, we represent abstract information sets by the latent vector and we use Euclidean distance to find the closest representative.
> > > 3. If we understand the question correctly, the reviewer is asking why do we learn $\Lambda_{i, \theta}^I$, if we always use $\Lambda_{i, \theta}$ to construct the whole public state. Please if this is not the answer to your question let us know so that we provide better answer.
> > > We generate all the combinations, but different actions can lead to the same public state, but different information set. That is why we learn the $\Lambda_{i, \theta}^I$, so that we are able to identify which of the abstract information sets corresponds to the one encountered in the game during test time.
> > > 4. The strategy $\pi_\theta$ is trained for the real information sets. The strategy form which we sample in Figure 7, is not this  $\pi_\theta$, but it is strategy computed in the look-ahead reasoning, which we extract from the subgame as described in answer to question 3. The $\pi_\theta$ is not used during the inference at all, LAMIR uses it only to guide the training of the value function.

---

> > > ### Author Response · Authors · 2025-12-03
> > >
> > > ## Regarding fairness
> > > 1. We understand the critique of the reviewer, but we argue that even if the LAMIR would not be able to outperform RNaD with the same train time, our main contribution is using additional test-time compute, which policy-gradient algorithms generally cannot do in imperfect information games.
> > > Still, we provide the result of this experiment. Due to time restraints we could not do any additional training. Instead, we have used the already trained networks. For RNaD we have used the same checkpoint as in the original experiments. For LAMIR we have used earlier checkpoint, which took roughly the same time to train as the final checkpoint of RNaD. For Goofspiel 10, this checkpoint was after 1.2M episodes, for Goofspiel 13 and 15 it was after 1.4M episodes. We have used only the LAMIR with legal actions used for clustering.
> > > * Goofspiel 10: $47.07 \pm 1.27$ %
> > > * Goofspiel 13: $54.14 \pm 1.16$ %
> > > * Goofspiel 15: $62.85 \pm 1.12$ %
> > >
> > > From our earlier experiments, we have trained RNaD far beyond the 100k iterations used in that experiments and we can safely say that RNaD does not significantly improve beyond the reported values even after 1M iterations, so even in that experiment with the same resources, the RNaD would not outperform LAMIR.
> > >
> > > 2. We have run the training for 1000 iterations training steps in Goofspiel 5. These are the times the training took:
> > > * RNaD: 48s
> > > * $L= 5$: 98s
> > > * $L=10$: 99s
> > > * $L=15$: 104s
> > > * $L=20$: 110s
> > > * $L=25$: 117s
> > > * $L=30$: 132s
> > >
> > > ## Regarding merging abstraction
> > > 1. We described this merging process in answer to Question 2 about Figure 7. To summarize: We compute Euclidean distance between each node and each representative of the current public state and then map each node to the closest one.
> > > 2. To clarify our previous answer. If the $L$ is greater than the largest public state and $\kappa$ is injective, then the soft clustering ensures that every abstract information set $\overline{s_i}$ corresponds to only at most one real information set $s_i$. If this is the case, then after the soft clustering converges, from that point onward the $\Gamma_\theta$ will always be trained on the same targets and as a result it will be eventually trained to perfectly construct the original game tree (but with the abstract information sets). If the $L$ is smaller, then the clustering also converges to some cluster centers and $\Gamma_\theta$ will learn to predict transitions between those centers. This is what we meant with “consistency should naturally emerge”. However, to the question whether all leaf nodes can be covered, the answer to this question is generally no, because decreasing the size of the public states to fixed amount is inherently lossy abstraction. As a result, there cannot exist an abstraction with fixed size $L$, where all leaf scenarios are adequately represented. Still our abstraction is learned on every possible trajectory, so each leaf node reached during training is represented in the abstraction, but this representation does not necessarily have to be adequate due to the fact that multiple leafs will be mapped to the same outcome. In the middle part of Figure 4, we construct the whole game tree, so there is no merging, still this experiment results in exploitability being on-par or worse than applying the whole LAMIR with merging.

---

### Official Review · Reviewer_eWxn · 2025-10-29

**Soundness:** 3
**Presentation:** 3
**Contribution:** 4
**Rating:** 8
**Confidence:** 4

**Summary:**

The authors propose abstract look-ahead search at test time for imperfect-information games. While training RNaD policy, they learn a world model over latent information sets. They then learn a per-public-state abstraction that clusters many real infosets into latent prototypes, yielding a depth-limited latent subgame that's smaller than the original game or world model. At the depth limit they attach a policy-transformation leaf gadget (from Kubíček et al., 2024) and solve the subgame with CFR. Test-time look-ahead achieves lower exploitability than vanilla RNaD in small games and beats RNaD in larger games.

**Strengths:**

- This paper makes important and large contributions towards using latent-representation world models effectively in a 2p0s imperfect information self-play setting.
- The proposed method is relatively straightforward.
- The experimental results are sufficient and convincing.
- The limitation to games without chance and lack of convergence grantees for games without A-loss recall are well explained.

**Weaknesses:**

- The proposed feature encoding for latent infostate clustering is a heuristic.
- Code is not provided.
- Minimal ablations on the world model architecture design.
- Missing related works: TD-MPC and TD-MPC2

**Questions:**

a) This method seems like it would be extendable to other self-play methods like magnetic mirror descent. Would you expect clear integration issues or challenges in using this lookahead method with population-based methods like PSRO?

b) Is any regularization or activation applied to shape or constrain the space of latent infostates $\bar{S}_i$?

c) Without any loss of novelty, the base world model dynamics and encoder model seem mechanically very similar to the one used in TD-MPC [1] and TD-MPC2 [2] (A key distinction is that LAMIR operates over information sets under partial observability.)
 Would the world model stability improvements introduced in TD-MPC2 potentially be applicable to this work?
e.g.
1. Only training the initial state encoding $\Lambda_\theta^I(s_i^{t})$ using dynamics loss and stopping the gradients through dynamics targets $\Lambda_\theta^I(s_i^{t+k})$
2. Bounding and regularizing the latent infostate space with SimNorm

[1] Hansen, N., Wang, X., & Su, H. (2022). Temporal Difference Learning for Model Predictive Control.ICML 2022 (PMLR).

[2] Hansen, N., Su, H., & Wang, X. (2024). TD-MPC2: Scalable, Robust World Models for Continuous Control.ICLR 2024.

**Details Of Ethics Concerns:**

No ethics concerns.

---

> ### Author Response · Authors · 2025-11-19
> **Rebuttal**
>
> We would like to thank the reviewer for their time and for pointing us to related work we have missed.
>
> “Not available code”: The code should be available in the supplementary zip in the folder lamir. If there are some parts of the code missing, please let us know.
>
> Question a) “Extension to MMD/PSRO” Thank you for pointing this out, it would indeed be possible, and we mention more details about it in the related work paragraph on Direct policy optimization. There are several ways the PSRO could be combined with our approach. One possibility is to use the same traces that PSRO uses for best-response computation in the training of our model. In addition, the best responses trained by PSRO could be used as a basis for our multi-valued states value function. We believe this may be an interesting direction for future work.
>
> Question b) “Is any regularization or activation applied to shape or constrain the space of latent infostates?” The only constraints placed on the abstract information sets are forced by the neural network architecture, i.e., each abstract information set is represented by a vector $mathbb{R}^S$, where $S$ is a hyperparameter. In the experiments, we have used MLP to represent the abstraction network, and we have not used any output layer activation or normalization. However, we have used ADAM with weight decay as our optimizer, which dampens the weights, so that the values do not blow up.
>
> Question c) “Similarity to TD-MPC and TD-MPC2” We would like to thank the reviewer for pointing out this missing work. It is highly relevant, and we have included it in the revision of the related work. We agree that the model is mechanically similar, except using information sets instead of states, and by using the architecture to bound the size of the public state, which is important in imperfect information reasoning. We believe that the SimNorm could be used off-the-shelf. As for the training $\Lambda_{\theta}^{I}(\Infoset^t_i)$ of only the initial state encoding (and not propagating it further with the dynamics function), we believe that this is already a thing we do. We have revised the loss in equation (3), which states this more clearly.

---

### Official Review · Reviewer_nqxi · 2025-11-04

**Soundness:** 3
**Presentation:** 3
**Contribution:** 3
**Rating:** 6
**Confidence:** 2

**Summary:**

This paper presents a new method for building an agent that plays two-player zero-sum imperfect-information games, using a lookahead search technique similar to MuZero.

The main Dynamics function maps a joint latent infostate and a joint action to a new joint latent infostate. It is learned through sampled trajectories via a simulator. It can then be used at test-time with depth-limited solving to play well.

Additionally, the method uses an abstract model to deal with large public states.

The paper includes experiments computing exact exploitability in small Goofspiel and Oshi-Zumo games, and head-to-heads results against RNaD in large Goofspiel.

**Strengths:**

The research direction and idea are sensible. The method is explained well. The experiments are solid.

**Weaknesses:**

I don't quite understand the Abstract Model (see questions).

I think more could be said about the motivation/application of this work. It's practically useful only if we (A) don't already have a perfect simulator but (B) have access to trajectories that include both players' infostates and actions. When could this be useful?

**Questions:**

1. Apologies if I missed it in the paper, but is it true that this method only works on games with no chance?
2. While playing the game, is it possible that the root public state of the game tree contains more than $L$ information states?
3. I'm a bit confused on Section 4, especially what it is used for. I think Section 4 describes how to train $\Lambda_{i,\theta}$, $\kappa_{\theta}$, and $\Lambda^I_{i, \theta}$.

However, the description of the second loss says that it updates $\Lambda^I_\theta$. Is this a typo? Should it be $\Lambda^I_{i, \theta}$? But the equation describing the second loss only includes $\Lambda_{i, \theta}.

In the description of LAMIR at the end of Section 5, these are only used to map the real current infoset $s_i$ to the abstracted one $\bar{s_i}$. Is this all it is used for? And why can't we just use $\Lambda^I_\theta$ to do this?
4. In Section 4, why do we need to map latent infostates and infostates to a K-dimensional space? Why can't we just do everything in the latent space?

---

> ### Author Response · Authors · 2025-11-19
> **Rebuttal**
>
> Thank you for pointing out the parts of the paper, which are not clear. We have revised section 5 to improve the overall clarity of LAMIR.
>
> Usefulness: We argue that our approach is useful not only when we do not have the perfect simulator, but also when we do. When we say “simulator”, we mean an environment that stores its internal state, and when passed a joint action, it modifies its internal state and provides observations. In training, this is precisely the simulator we use to generate traces. Such a simulator cannot be used for search even in perfect information games, as it only allows going forward in the trajectory (e.g., we cannot start from the same non-initial state). This is a standard Reinforcement learning setting.
>
> On the other hand, the model of the game we learn is a stronger notion, as it allows executing all available joint actions in a single state and observing all possible observations and rewards. In our case, the model is also automatically abstracted to maintain a small constant size. The sound look-ahead reasoning in imperfect information games must be initiated from several states that share the common knowledge. In games like Dark Chess, Stratego, and Imperfect Information Goofspiel 15, there may be over 10^15 of such states, making sound reasoning intractable even with the perfect model. The goal of the approach is to learn an abstraction with bounded size that will enable this look-ahead reasoning regardless of the size and complexity of the simulator.
>
> Question 1: “Is it true that this method only works on games with no chance?” Similar to the initial MuZero paper, we currently assume only games without chance; examples of such games are Dark Chess, Stratego, and Battleships. Adding chance is possible, but it introduces additional technical challenges and increased sample complexity. We would like to explore the extension to games with chance in future work.
>
> Question 2: “Is it possible that the root public state of the game tree contains more than L information sets?” It is possible in the original game. In the abstract model $\Lambda_{i, \theta}$, the neural network architecture puts a hard constraint on each public state that it consists of at most $L$ abstract information sets. Since the root of any subgame is a public state, it can never happen that there are more than $L$ information sets within the root, which results in at most $L^2$ possible abstract world states.
>
> Question 4: “Why can't we just do everything in the latent space?” It is possible to do the clustering in the latent space. However, the goal of the clusters is to map similar information sets to the same abstract representative. The $\kappa$ is used to provide information about this similarity.

---

> ### Author Response · Authors · 2025-11-19
> **Answer to question 3**
>
> Question 3: “Role of \Lambda_{i, \theta}^{I}” Section 4 introduces the abstract model we use and then introduces a loss function to train such a model. We have revised Section 5, so that it contains this explanation of the whole training procedure better. For completeness, we will also provide a similar explanation below. The individual parts of the abstract model are:
>
> 1. $\Lambda_{i, \theta}$ - Network which maps the public state representation to an $L$ abstract information set representations of player $i$ (which is the hard constraint from question 2 on the size of the public state). In the experiments we have used $\Lambda_{i, \theta}: \mathbb{R}^{S_0} \to \mathbb{R}^{L \times S}$, where $S_0$ is the size of public state representation (e.g. the vector which stores all public information), $L$ is the abstraction size and $S$ is the size of the abstract information set latent representation. Both $L, S$ are hyperparameters.
>
> 2. $\Lambda_{i, \theta}^I$ - Network that maps the real information set representation to the distribution over the $L$ abstract information sets produced by $\Lambda_{i, \theta}$. One possible implementation is $\Lambda_{i, \theta}^I: \mathbb{R}^{S_i} \to \mathbb{R}^{L}$, where $S_i$ is the size of the information set representation (e.g. the vector that stores all private information of the player $i$). The output is a probability distribution over the abstract information sets from $\Lambda_{i, \theta}$.
>
> 3. $\kappa_{\theta}$ - Network which maps the abstract information set to the $K$-dimensional space in which we cluster. Here $\kappa_{\theta}^I: \mathbb{R}^{S} \to \mathbb{R}^{K}$, where $K$ depends on the chosen property (for legal actions or strategy $K = |\mathcal{A}|$)
>
> As for the loss functions, the first one $\mathcal{L}^A$ performs the soft clustering in the $K$-dimensional space represented by $\kappa$. This is used to train the $L$ representatives. The second one $\mathcal{L}^S$ is a cross-entropy between the probability distribution produced by $\Lambda_{i, \theta}^I$ and the closest representative from $\Lambda_{i, \theta}$. Assume a single training example for information set $s_{i}$ that is contained in public state $s_{0}$. First the public state is mapped through $\Lambda_{i, \theta}$ to $L$ abstract information sets $\overline s_{i, l}$, $l \in \{1, 2, \dots, L\}$. We will use legal actions as $\kappa$, then each $\overline s_{i, l}$ is mapped to the  $|\mathcal{A}|$-dimensional space as $A_{i, l} = \kappa_{\theta}(\overline s_{i, l})$, where $A_{i, l}$ is a vector representing whether each possible action is legal or not. Note that $A_{i, l}$ does not need to be a binary vector, but it is just a vector in $\mathbb{R}^{|\mathcal{A}|}$. The real information set $s_{i}$ has legal actions $\mathcal{A}(s_{i}) = \kappa(s_i)$, which is a binary vector in $\mathbb{R}^{|\mathcal{A}|}$. The soft clustering computes the distance between $\mathcal{A}(s_{i})$ and each $A_{i, l}$ and uses the weighted sum of these distances as a loss function (equation (2)). The weight is the softmax of the negative distances. This loss propagates through  $\kappa_{\theta}$ and $\Lambda_{i, \theta}$. There is a single abstract information set $\overline s_{i, B} = argmax_{s_{i, l}} ||\kappa_{\theta} (\overline s_{i, l}), \kappa (s_i)|| $. The  $\Lambda_{i, \theta}^I$ maps the $s_{i}$ to a probability distribution over the $\overline s_{i, l}$. The second loss in equation (3) trains the $\Lambda_{i, \theta}^I$ to predict the $\overline s_{i, B}$ with probability 1 and the rest with probability 0.
>
> $\Lambda_{i, \theta}^I$ is used in equation 3, it is used to map the $s_{i}$ to a probability distribution before the loss is computed. We thank the reviewer for pointing out that this part was not clear. We have revised equation (3) and the explanation of the entire method in Section 5 to improve clarity. We would like to ask whether these changes have made the concepts clearer
>
> It is true that $\Lambda_{i, \theta}^I$ is used purely to map the real information set to the abstract one (or more precisely to the distribution over the abstract information sets). This is necessary because the player needs to know which strategy it should use after it solves the subgame, because there will be $L$ strategies, one for each abstract information set. The $\Lambda_{i, \theta}^I$ then identifies which of the abstract information sets correspond to the real one the best. It is possible to train $\Lambda_{i, \theta}^I$ so that it directly predicts the latent representation of the abstract information set and not the distribution. However, this would create additional dependency as the latent representation produced by $\Lambda_{i, \theta}^I$ would need to be consistent with the latent representation produced by $\Lambda_{i, \theta}$. Using a probability distribution instead of directly predicting the latent representation avoids this issue as it always uses the latent representation from $\Lambda_{i, \theta}$.

---

> > ### Comment · Reviewer_nqxi · 2025-11-19
> >
> > Thanks for the response.
> >
> > I will read this and try to understand.
> >
> > I'll initially address one point though -- I'm not sure if you have addressed one of the questions I asked in Question 3:
> >
> > On ~line 265, it's noted that the second loss updates $\Lambda^I_\theta$. However, this term is not otherwise mentioned at all in Section 4 or in your explanation here -- it is only used in Section 3. Is it a typo? If not, can you expand more on the role of Section 4 in training $\Lambda^I_\theta$?

---

> > > ### Author Response · Authors · 2025-11-19
> > >
> > > We apologize for this, it was an oversight. In section 3 we have defined  $\Lambda^I_{\theta}$ as mapping from real information set to latent representation. From section 4 onward, we replace this with $\Lambda^I_{i, \theta}$ and $\Lambda_{i, \theta}$, which should correspond to the abstractions. Using $\Lambda^I_{\theta}$ was just a typo and we have replaced it in new revision. Just as an additional note, the subscript $i$ represents to which player the network corresponds to.

---

### Author Response · Authors · 2025-12-03
**Summary of the changes made throughout the rebuttal stage.**

We would like to thank all the reviewers once again for their reviews, suggestions, and discussion during the review period. Their notes helped us improve the quality of the paper. We will now summarize the main points from the whole process.

**Motivation:**  All the reviewers agreed that the motivation of the paper is sensible and important. Moreover, the reviewers seem to agree that the experiments are solid as Regularized Nash Dynamics (RNaD) which we have used as a baseline for comparison is a strong algorithm and our approach managed to outperform it both in small and large games. One reviewer questioned the usefulness of the approach as we require a perfect simulator with both players actions and information sets. We argue that our approach is useful even with a perfect simulator as it limits the size of the subgame to be tractable for look-ahead reasoning

**Clarity:** Amongst the reviewers, there were also some concerns about the presentation of the paper. We have used the additional page available during the rebuttal to revise Section 5, in which we describe both the training and testing process in more detail with the individual components. We have also added schemas and pseudocode into the Appendix, which helps to further enhance the clarity.

**Fairness of experiments:** There was a reasonable concern by one reviewer, that the proposed algorithm, LAMIR, takes more training resources compared to RNaD and in our experiments we have compared those two algorithms with the same number of iterations and not the same training time. We have run the experiment which compares the two algorithms with roughly the same training time. As expected LAMIRs performance deteriorated, but still it managed to outperform RNaD in the larger instances.

**Size of games in experiments:** Some reviewers also mentioned that LAMIR is devised for large games, but in experiments we do not use complex enough games so that they do not need the learning of the model. We think this is due to underestimation of the games we use. Those games have simple rules, yet their size grows very quickly. In games used in Section 6.2, the traditional look-ahead techniques without abstraction cannot be applied. We have added the table highlighting their size to the Appendix. As one of the reviewers noted,  it may be possible to hand-make abstractions to make the used games smaller, but the motivation of LAMIR is to find these abstractions automatically in self-play, just from the training data.

**Novelty:** One reviewer questions the novelty of the approach. It is true that our approach builds upon several existing  techniques, but it combines them in a non-trivial way. Moreover, we believe that the usage of soft clustering in self-play to find the abstraction to keep the public states tractable is a novel contribution of the paper.

Next we summarize the changes we have made during the review process:
* Expanded the related work with suggestions made by the reviewers.
* Expanded and revised Section 5, so it contains the clearer description of both the training process and the reasoning.
* Changed equation (3) such that it clearly shows which network is trained.
* Included in Section 6.1 the explanation of the training time of LAMIR compared to RNaD.
* Fixed the typos in the main text and mistakes in used symbols.
* Added Algorithm 1 and Figures 7 and 8 to the Appendix which summarize both the training and the test-time reasoning.
* Changed Figure 1 so it is consistent with added Figure 7.
* Added new experiment into Appendix that shows the exploitability of the strategy for larger depth limits
* Added a Table to Appendix detailing the effect of increasing abstraction size to training time
* Added an experiment to Appendix comparing LAMIR and RNaD given the same training time.
* Added a Table to Appendix that contains the approximate sizes of the games used in experiments.

---

### Meta-Review · Area_Chair_iikK · 2025-12-24

**Summary:**

The reviewers generally agreed that this paper addresses an important and timely problem: enabling principled look-ahead reasoning in large imperfect-information games without access to an explicit game model. The proposed approach, LAMIR, combines learned world models, online abstraction of information sets, and depth-limited subgame solving, and was viewed as technically sound with solid empirical results.

The main concerns raised across reviews focused on (i) clarity of presentation, particularly regarding the training and test-time interaction between abstraction, model learning, and continual resolving; (ii) the computational fairness of comparisons with RNaD, given LAMIR’s higher per-iteration training cost; (iii) the novelty of the approach relative to prior work on abstraction, subgame solving, and learned dynamics; Reviewers also questioned whether the experimental domains sufficiently demonstrated the necessity of learned models and abstractions, and whether deeper search or more complex games would expose potential failure modes.

Overall, while reviewers acknowledged that the method builds upon existing components, most agreed that their integration in the imperfect-information setting is non-trivial, and that the empirical evidence supports the paper’s core claims.

**Reviewer Concerns:**

Concerns addressed by the rebuttal:

•	Clarity and algorithm description:

Multiple reviewers found Section 5 and the interaction between components difficult to follow. The authors responded by revising Section 5, correcting notation errors, and adding detailed pseudocode and figures in the appendix. Follow-up comments from reviewers explicitly acknowledged that these additions improved understanding.

•	Training cost and fairness of comparison:

Reviewers raised reasonable concerns that LAMIR is more expensive per iteration than RNaD. The authors clarified the source of the overhead, reported per-iteration slowdowns, added tables showing how cost scales with abstraction size, and provided additional experiments comparing LAMIR and RNaD under roughly matched training time. While LAMIR’s advantage narrows under equal wall-clock time, it remains competitive or superior in larger games, which adequately addresses the fairness concern.

•	Experimental scale and necessity of abstraction:

Several reviewers questioned whether the evaluated games were sufficiently complex. The authors clarified the size of the public state spaces involved and added tables quantifying game sizes, strengthening the argument that abstraction is essential in these domains.

•	Technical questions on abstraction, clustering, and merging:

Detailed questions about the role of abstraction networks, clustering losses, and merging behavior during search were answered in the rebuttal, including explicit acknowledgment of abstraction-induced lossiness and its implications. The authors were transparent about the lack of strong theoretical guarantees and clarified when and why abstraction errors may arise.

Concerns that remain partially outstanding:

•	Novelty relative to prior work:

While the rebuttal explains how the method combines existing ideas in a non-trivial way, the contribution is still best characterized as an integration rather than a fundamentally new algorithmic primitive. This concern remains a matter of perspective rather than a clear flaw.

•	Generality beyond current experimental domains:

The method is not evaluated on domains such as poker or games with chance nodes, and deeper search sometimes degrades performance when abstractions are small. The authors acknowledge these limitations and frame them as directions for future work, but they remain open questions.

**Reviewer Scores:**

•	Reviewer nqxi (initial score: 6)
This reviewer’s main concerns centered on clarity, notation issues, and the role of abstraction components. These were explicitly addressed through revisions, corrections, and detailed explanations. Given the reviewer’s own low confidence and their acknowledgment of improved clarity, it is likely their score would increase slightly.

•	Reviewer eWxn (initial score: 8)
This reviewer was already strongly positive and raised mainly minor concerns (missing related work, limited ablations). These were addressed in the rebuttal. The score would likely remain 8.


•	Reviewer q3UX (initial score: 6)
This reviewer had extensive concerns about clarity, computational cost, abstraction consistency, and experimental scope. The authors made efforts to respond, added experiments, and were transparent about limitations. While some skepticism about abstraction quality and deeper search remains, the reviewer acknowledged improved understanding. The score would likely increase modestly.

•	Reviewer MHic (initial score: 6)
This reviewer questioned novelty and experimental domains but explicitly stated they would maintain their rating after the rebuttal. The score would likely remain 6.

Overall Assessment

The paper presents a technically sound and well-motivated approach to a challenging problem in imperfect-information games. While not all limitations are resolved, the authors respond constructively and transparently to reviewer concerns, and the empirical results support the paper’s main claims. On balance, the strengths outweigh the remaining weaknesses, and I recommend acceptance.

---

### Decision · Program_Chairs · 2026-01-26

Accept (Poster)